# Real-Time Artifacts Reduction during TMS-EEG Co-Registration: A Comprehensive Review on Technologies and Procedures

**DOI:** 10.3390/s21020637

**Published:** 2021-01-18

**Authors:** Giuseppe Varone, Zain Hussain, Zakariya Sheikh, Adam Howard, Wadii Boulila, Mufti Mahmud, Newton Howard, Francesco Carlo Morabito, Amir Hussain

**Affiliations:** 1Department of Medical and Surgical Sciences, Magna Greacia University of Catanzaro, 88100 Catanzaro, Italy; giuseppe.varone1@studenti.unicz.it; 2College of Medicine and Veterinary Medicine, University of Edinburgh, Edinburgh EH16 4TJ, UK; zain.hussain@ed.ac.uk (Z.H.); z.sheikh-1@sms.ed.ac.uk (Z.S.); 3Howard Brain Sciences Foundation, Providence, RI 02906, USA; howard.adam@mayo.edu; 4RIADI Laboratory, National School of Computer Sciences, University of Manouba, Manouba 2010, Tunisia; wadii.boulila@riadi.rnu.tn; 5IS Department, College of Computer Science and Engineering, Taibah University, Medina 42353, Saudi Arabia; 6Department of Computer Science and Medical Technology Innovation Facility, Nottingham Trent University, Clifton, Nottingham NG11 8NS, UK; mufti.mahmud@ntu.ac.uk; 7Nuffield Department of Surgical Sciences, University of Oxford, Oxford OX3 9DU, UK; newton.howard@nds.ox.ac.uk; 8DICEAM Department, “Mediterranea” University, I-89122 Reggio Calabria, Italy; 9School of Computing, Edinburgh Napier University, Edinburgh EH11 4BN, UK; a.hussain@napier.ac.uk

**Keywords:** transcranial magnetic stimulation (TMS), electroencephalography (EEG), TMS-EEG, TMS-artifacts, EEG amplifier and headset, TMS-Evoked potential (TEPs), synchronization tools, TMS-EEG laboratory layout, subject preparation, online tricks for TMS artifact minimization

## Abstract

Transcranial magnetic stimulation (TMS) excites neurons in the cortex, and neural activity can be simultaneously recorded using electroencephalography (EEG). However, TMS-evoked EEG potentials (TEPs) do not only reflect transcranial neural stimulation as they can be contaminated by artifacts. Over the last two decades, significant developments in EEG amplifiers, TMS-compatible technology, customized hardware and open source software have enabled researchers to develop approaches which can substantially reduce TMS-induced artifacts. In TMS-EEG experiments, various physiological and external occurrences have been identified and attempts have been made to minimize or remove them using online techniques. Despite these advances, technological issues and methodological constraints prevent straightforward recordings of early TEPs components. To the best of our knowledge, there is no review on both TMS-EEG artifacts and EEG technologies in the literature to-date. Our survey aims to provide an overview of research studies in this field over the last 40 years. We review TMS-EEG artifacts, their sources and their waveforms and present the state-of-the-art in EEG technologies and front-end characteristics. We also propose a synchronization toolbox for TMS-EEG laboratories. We then review subject preparation frameworks and online artifacts reduction maneuvers for improving data acquisition and conclude by outlining open challenges and future research directions in the field.

## 1. Introduction

Transcranial magnetic stimulation (TMS) is a non-invasive form of brain stimulation which uses a strong magnetic field to stimulate specific areas of the brain [1]. It differs from peripheral stimulation, which results in motor evoked potentials (MEPs) and somatosensory-evoked potentials (SEPs), as it can bypass sensory pathways and subcortical layers. TMS elicits an electroencephalography (EEG) response and the TMS-evoked potential (TEPs) can be recorded and serve as a reflection of cortical reactivity to TMS. A single pulse applied to the motor cortex leads to a localized and strong response at the site of stimulation. Neuronal excitement spreads from the motor area to ipsilateral premotor areas within 5 milliseconds (ms) and there is an activation of contralateral homologous cortical areas within 20 ms [2,3,4,5,6]. The use of TMS along with EEG (termed TMS-EEG) allows us to externally examine brain states, including their phase dynamics across motor and non-motor cortical areas [7,8]. The approach is capable of recording the time taken to resolve cortico-cortical interactions to within ms [9] in both normal and pathological brains [10] and has provided insights into excitatory and inhibitory human brain mechanisms [11,12]. It is also a powerful tool for assessing cortical dynamics at rest and during tasks [13,14].

TMS can be applied in a wide range of paradigms [15], and consists of magnetic pulses delivered by placing an electromagnetic coil on a subject’s scalp. Magnetic pulses can either be delivered in isolation, termed single-pulse TMS (sp-TMS), or in rapid sequences, termed paired-pulse TMS (pp-TMS). sp-TMS and pp-TMS are usually used to probe and measure cortical excitability in response to single or transient pulses.

However, effective paradigms which study cognition or brain disorders generally use repetitive TMS (rTMS), which has longer lasting effects by introducing neural plasticity effects that persist after the stimulation. rTMS can inhibit or decrease excitability based on the intensity of the stimulation, with low frequency rTMS (≤1 Hz) inhibiting cortical firing and high frequency (≥5 Hz) provoking firing. A novel patterned form of rTMS is theta burst stimulation (TBS), which usually consists of three bursts of pulses given at 50 Hz and repeated every 200 ms, and can be applied using continuous or intermittent protocols. In continuous TBS (cTBS), the pattern persists for 40 second (s) and is uninterrupted, whereas in intermittent TBS (iTBS), short patterns (e.g., 2 s) are spaced with a rest time (e.g., 8 s). iTBS increases human cortical excitability whilst cTBS has the opposite effect. TBS can be used to index brain plasticity and is a powerful tool for exploring local cortical and brain network plasticity. There are also a number of TMS paradigms that are assumed to work over similar physiological cortical properties, which include Quadri-Pulse Stimulation (QPS), transcranial Alternating Current Stimulation (tACS), Paired Associative Stimulation (PAS), controllable pulse shape TMS (cTMS) and deep-brain TMS (dTMS) [16,17,18,19,20,21]. These paradigms are usually designed to assess cortical feed forward propriety, instantaneous status, intrinsic oscillatory activity, connectivity and event related response phase dynamics [2,22,23,24].

It is evident from the current literature that TMS has great potential in providing novel insights into the pathophysiology of neurological and psychiatric disorders. It can also facilitate studies into the relationships between cognition and behavior and has both diagnostic and therapeutic potential for various neurological disorders [16,25,26,27,28,29,30,31,32]. Electrical activity in the brain can be transduced using scalp EEG sensors, which represent the summation of postsynaptic potentials of underlying pyramidal neurons [33], and provide a non-invasive approach for measuring TEPs. High density EEG is a technique which uses 32–256 electrodes to record electrical brain activity with high-temporal resolution [34] and good spatial localization. The first TMS-EEG experiment was carried out by Cracco et al. [35], and since then, a number of challenges in recording EEGs for TMS have been identified. Several years later, Ilmoniemi et al. [2] were able to demonstrate for the first time that TMS with a high-resolution EEG can be used to not only measure early TEPs, but to also quantify and characterize the spread of the activation across time, spatial and frequency domains. However, due to the nature of TMS, different artifacts can disrupt the ongoing neuronal activity and can mask the natural TEPs. The resulting high voltages and long-lasting components make it difficult and computationally expensive to disentangle early TEP components (≤20 ms) [2,36,37].

In addition, somatosensory and auditory artifacts also arise as the TMS pulse is often associated with a loud clicking sound, which can stimulate peripheral sensory and motor axons. Artifacts can also be induced due to electrode movement and polarization, eye movements, muscle activation and the coil touch. A number of strategies have been proposed and applied in an attempt to address the challenges associated with disentangling noise components from TMS and to maximize the signal-to-noise ratio of stereotypical evoked brain responses [23,37,38,39,40,41,42]. These have had varying degrees of success and so this still remains an open challenge.

In the literature, there are a limited number of papers which survey artifacts in EEG based readings of TMS-evoked responses. For example, Rogasch et al. [43] and Farzan et al. [44] reviewed the origins of such artifacts and methods for removing them. Whilst they both discussed techniques for preventing and correcting artifacts, they did not review and explore the role of EEG technologies. Another comprehensive review in the field is by Tremblay et al. [45]; however, its primary focus was on the clinical utility and prospective of TMS-EEG.

Our survey aims to provide an overview of studies in the TMS-EEG field over the last 40 years. We review the sub-types and causes of TMS-EEG artifacts and present the state-of-the-art in EEG technologies. We then propose a synchronization toolbox for TMS-EEG laboratories. We also identify methodological challenges in conducting high quality experiments, outline solutions to overcome these and discuss open challenges in the field.

The rest of this paper is organized as follows: Section 2 presents our methodology; Section 3 depicts electrical and equipment related TMS-EEG artifacts; Section 4 presents TMS-coil related artifacts; Section 5 reports technologies that deal with magnetic artifacts; Section 6 proposes a tool to synchronize TMS laboratory equipment; Section 7 details subject preparation steps to improve TEPs acquisition; Section 8 explores existing challenges and future work and, finally, Section 9 concludes our survey.

## 2. Research Methodology

To identify relevant research topics in the TMS-EEG field, a literature search was conducted using Scopus and Google Scholar. We only included articles written in English and published in peer-reviewed academic journals. We searched using the following keywords: transcranial magnetic stimulation (TMS), artifacts, EEG amplifier technologies, TMS technical and methodological improvements, TMS-EEG equipment, TMS-EEG review, TMS-EEG principles, TMS coil and neuronavigation systems. Reference lists of all selected articles were also reviewed to locate any other relevant publications not found in the original search. Initial screening was performed on titles and abstracts and the included publications were subsequently screened using the articles’ full texts. After full-text screening, we identified over 300 relevant studies related to TMS-EEG research. To refine our search, we only included papers which discussed experiments and tests related to: (i) TMS coil shape; (ii) EEG sensors; (iii) TMS stimulator units; (iv) Pulse length; (v) Recharge artifacts; (vi) EEG headsets; (vii) EEG amplifier front and back-end improvements; (viii) TMS-EEG artifacts and electrophysiological characterization (e.g., latencies, frequencies, timing, voltage, number of components); and (ix) Frameworks addressing subject preparation and artifact reduction. We excluded all publications in books and conference proceedings, abstract-only articles and articles on TMS for drugs and rehabilitation applications.

## 3. Electrical and Equipment Related TMS Artifacts

In the following sub-sections, we focus on artifacts that are associated with TMS stimulator units, EEG headsets and electrodes.

### 3.1. TMS Pulse Artifact

The application of a time-varying magnetic field of 2–3 Tesla lasting ∼200 µs induces an electric field of ∼100 mV/mm, which can result in large EEG spikes several orders of magnitude larger than normal neural activity [43]. This is termed the TMS pulse artifact. According to [23], the TMS pulse artifact typically lasts for ∼5 ms and the amplitude of it can be reduced by changing the maximum simulator output intensity (%MSO). In Table 1, we review the state of the art in online methods for minimizing artifacts and report tools and tips commonly used in TMS-EEG experiments.

The source of this artifact remains unclear but it has been hypothesized to be related to electrode polarization [38], slow capacitive effect at the sensor/skin interface [72], muscle activity polarization [22,28,37,83,85], strong and long-lasting cortical potentials [36] and with the electrical equipment used [23]. The artifact also varies based on %MSO, pulse length, frequency and waveform, coil shape and Magnetic Stimulator type [86]. Commonly employed procedures to minimize this artifact include good impedance (<5 KΩ) [72], orienting the wires of the sensors arranged orthogonal to the coil handle [38] and the use of insulated sensors, shielding wires [54], and thin Ag/AgCl sensors [55] with C-shape or pellet shape. In addition, Cl^−^ electrolytic paste can help prevent slow capacitive discharging effect [47].

### 3.2. TMS Recharge Artifact

TMS units use a large capacitor, storing several kilovolts to enable discharge in the coil within ∼100 microseconds (µs). During rTMS applications, TMS capacitor performance depends on parameters such as the intensity and pulse frequency. Thus, for a short Inter-Pulse-Interval (IPI), the discharge time will be faster than the recharge time. To avoid capacitor performance reduction, the TMS capacitor starts to recharge before the previous pulse energy expires. In this circumstance, the capacitor recharge energizes the “tail” of the dissipation of the previous pulse. In other words, the recharge artifact is due to the amount of recharge current during the rapid energy-restoring in the capacitor. This phenomenon is modelled by an exponential function. In general, the electrodes near the hotspot site are the ones most contaminated by the charging artifact. It was found that in Magstim Super Rapid and Rapid2, two TMS simulators, the charge artifact on the EEG signal had a constant amplitude (12 µV) and latency as a threshold of %MSO [23]. In biphasic TMS units, the recharge delay can be manually set. As reported in [23], a successful approach to mitigate for the TMS recharge artifact is to manually set the recharge delay parameter on the TMS unit to an appropriate value (e.g., 1000 ms).

### 3.3. Electrode Motion

Electrode motion is generally caused by the accumulation of charge due to finger tapping, physical contact with the coil, air pressure changes or magnetic coupling. For example, finger tapping may disturb the electric distribution of charge at the skin–electrode interface, producing an artifact [37]. In [66], the authors report that the electrode movement artifact is primarily due to skin stretching or electrode contact that may result in a potential change of 10 mV. These artifacts result from a combination of parasite currents [23], additional charges at the skin–electrode interface [52] and electromotive forces at electrodes [38].

A number of strategies have been employed to minimize this artifact. Sekiguchi et al. [38] suggest ensuring that the TMS coil handle is perpendicular to the sensor wires and [53,54,55] suggest using insulated C-Shape or pellet shaped Ag/AgCl sensors. In [66], the authors report that scrubbing the skin reduced the artifact by two orders of magnitude. It is also important that any coil–sensor contact or finger contact with the sensors should be minimized to avoid this artifact [76,87]. Parasite currents at the electrolyte–paste–sensor interface can also lead to motion and heating during TMS [36,39] with a risk of burning the subject’s skin [88]. To avoid this, C-shaped or pellet Ag/AgCl sensors can prevent skin burns [39]. Plastic or Teflon insulated sensors [42,53] may also reduce TMS artifact coupling.

### 3.4. Electrode Polarization

Generally, EEG sensors can be categorized as either polarizable or non-polarizable. In polarizable sensors, the electrode–gel interface mimics a capacitor plate as the charge jumps from the skin to the electrode. Cl^−^ conductive gel creates a path between the skin and the transducer, ensuring a continuous flow of ionic current. Cl^−^ conductive gel can penetrate hair, settle on the surface of the scalp, coat the high impedance head skin stratum and infiltrate the inner layer of skin by passing through sweat glands and pores. During TMS pulse at the electrode–gel interface, polarization occurs. The capacitive electrode–gel interface can be charged via induction of currents in the wires and electrodes [38]. The electrode–gel interface resembles an electrolytic capacitor. The skin of the scalp consists of two external layer of epidermis, underneath which is a layer of dermis and then a layer of subcutaneous tissue. The epidermis plays a pivotal role at the electrode–skin interface and is formed of layers of new and old cells.

The dead cells form a layer called the stratum corneum, which is ten or more micrometers thick. The external space by the epidermal cell contains a fluid mixture. The stratum corneum acts as a capacitor with a hydrophobic dielectric but is usually interrupted by sweat glands and hair follicles. The epidermis layer at the gel–skin interface mimics a bi-dimensional array of capacitors with occasional small leakage, mimicking resistors. The slow recovery artifact at the gel–skin interface is depicted as a lumped model in [55]. The slow capacitive dissipation shows an exponential curve with a time constant gel and sensor characteristic dependent that takes hundreds of milliseconds to return to equilibrium. This slow discharging effect of charge dissipation is also known as decay effect [52]. The slow artifacts at the sensors–skin interface lead to capacitive discharging like a power law in time rather than the commonly assumed exponential. [55] with artificial deflection in EEG time series [89]. Julkunen et al. [72] report that electrode polarization is due to a capacitive effect at the electrode–electrolyte interface, resistance of the skin and parasite currents inducted by TMS in close ring sensors [41]. In [3,37,41,52], the authors report helpful tips to minimize this artifact, primarily aiming to reduce sensors impedance to <5 kΩ. Slow capacitive discharging usually occurs at electrodes close to the hotspot site. Scrubbing the skin where the electrodes are to be placed with an abrasive paste to remove granules of dead skin can help to improve impedance. Low-sensor impedance significantly reduces the amplitude and duration of the artifact [54,72]. It was also found that twisted and shielding wires, as well as plastic or Teflon insulated sensors, lead to reduced TMS artifact coupling [38,42,54]. However, this often does not turn out to be sufficient to reduce the effect of this artifact.

## 4. Muscle Activation and Spurious Potentials Evoked by TMS Coil

In this section, we present in detail the muscle artifact, blink artifact and spurious artifacts evoked by a TMS coil.

### 4.1. Muscle Artifact

In recent decades, several TMS-EEG studies have explored stimulation sites predominantly around the central sulcus [13,72] or the dorsolateral prefrontal cortex (DLPC) [90,91]. TMS activates the cells under the hotspot site and nearby tissues more. When TMS is administered at sites near the neck or forehead, this can directly depolarize the facial/scalp muscles or activate them by stimulating the relevant motor neurons [3]. This results in a muscle artifact due to the muscle fiber depolarization, which often presents with a biphasic signal with peaks at 4–5 ms and 7–10 ms and a very large magnitude (10 s of mV) detected by nearby electrodes [49]. Most commonly affected are the neck, jaw, facial [92], frontal, temporal and masseter muscles, depending on the position of the TMS coil [51]. Muscle artifacts appear as high frequency signals occurring in short bursts with high impacts in EEG signals [65]. If appropriate tools for TEPs analysis are used, the muscle artifact can be detected in real-time and reduced by reorienting the coil [13,38,46,49,61], reducing TMS intensity [62,93] or both. In [46,49,62], it was found that the muscle artifact was consistent with a similar M-wave pattern across recordings from peripheral muscles when the amplitudes were higher than 1 mV. These artifacts often last less than 50 ms but will still overlap and so mask the early natural TEPs components.

Procedures to minimize this artifact include re-orientating the sensor wires or the TMS coil handle [38,51], using shielding wire and insulated sensors [54], placing a thin layer of foam between the coil and hotspot site [13] and ensuring low impedance [23,66]. An appropriate setting of %MSO may also reduce muscle polarization and artifact contamination.

### 4.2. Eye Movement and Blink Artifact

TMS pulses can also trigger strong artifacts due to eye movement or blinking. Eye movements typically show a waveform with a high voltage that masks the natural TEPs response [94]. Across each eyeball, there is a cornea-retinal potential of 10–30 mV, which is larger than basal EEG activity. To reduce this artifact, different solutions have been proposed, including the use of an electro-oculogram (EOG) to monitor eye movements, instructing the subject to stare at a fixed point, soundproofing the subject during TMS experiments and using a thin layer of foam between the TMS coil and the hotspot sensor. Furthermore, when the blinks are synchronous with the TMS pulse, these are overlapped with the first components of the TEPs and, consequently, it is difficult to analyze the TEPs during the analysis of the TMS-EEG data [49,68].

### 4.3. TMS Confounding Factors: Coil Click and Somatic Sensation

A TMS pulse is usually accompanied by a loud click sound with a pressure of 100–120 dB [78,95,96] due to the electrical discharge of the TMS coil. This noise can trigger a peripheral auditory evoked potential (AEP) [70]. This noise and the corresponding response tone are particularly problematic in event-related potential (ERP) studies [37,41,70]. In [97], it was found that coil noise clicks elicit a positive response at 150 ± 250 ms post-TMS [97]. In more detail, it was noted that the noise clicks response lasts about 100 ms, with a prominent peak at 170 ms post-TMS stimulus. The AEP’s waveform consists of several peaks [98] with long components in TEPs such as P50, N100 and P180. In [70,97], it has been shown that N100 and P180 are correlated with auditory stimuli. In [70,75], it was found that AEPs may be conducted via air to the bone [73,99], reaching the cochlea directly and bypassing the middle ear. In addition, scalp stimulation can elicit a somatosensory response via trigeminal nerve activation [37,100], resulting in a Somatosensory Evoked Potentials (SEPs) [101]. SEP waveforms have their largest peak-to-peak amplitude (<4 µV) in the first 80 ms [100,102], and later peaks at (±200 ms) are smaller than (2 µV). However, the SEPs cannot be easily evaluated during the first (100–200 ms) as they can overlap with the muscle response [103].

To reduce TEPs noise, a number of different methods have been used. Earplugs [72,87] or earmuffs [77] alone cannot prevent the transmission of the “click”, but they may be used in tandem with other noise-masking procedures to effectively minimize auditory co-stimulation. Some researchers have also reported the successful use of white noise [36,75,80] or customized noise with the same reshuffle bins of frequency of the coil tone. Other researchers have merged white noise with synthesized noise to create “pink” noise, whose intensity was set at different levels attuned to each participant and used simultaneously with the TMS pulse to mask the noise [3,13,72,74,79,80,97,104]. These noise-masking procedures can substantially reduce the auditory-evoked artifacts, but no techniques can reduce the clicking noise to below audible decibel levels and low-frequency components are still often transmitted through bone conduction [73,99]. Using a thin layer of foam (∼0.5 cm thick) between the TMS coil and skin can reduce bone conduction as well as attenuate related somatic sensations and auditory confounding factors across the scalp [13,69,71,75,82,84,85], but the efficacy of this method is variable between subjects [73]. However, if the TEPs are symmetric [77,103,105], the AEPs are asymmetric and so the largest magnitude is over the contralateral hemisphere [100,106].

### 4.4. Artifacts Related with Parameter Setting and External Interference

In TMS-EEG experiments, artifacts in EEGs may arise even when the TMS stimulator is off, hence they are not always TMS and laboratory equipment related. These may include electrical and electromagnetic interference, subject discomfort, background sound and brightly lit rooms. Examples of electrical and electromagnetic interference that may produce an artifact include mobile phones, radio broadcasts, power lines (50/60 Hz), antennae, air conditioning, etc. A simple solution is to keep all these items away from the study equipment, by means of grounding arrangements or by a Faraday cage. However, there are a number of other factors which can produce artifacts. These include natural cardiac and respiratory patterns, and filtering tools to remove noise can produce new artifacts. For example, online notch filters used to remove 50–60 Hz noise can also produce “ripple” artifacts in the time series being analyzed.

## 5. Technologies to Deal with Strong Magnetic Artifacts

In this section, we survey state-of-the-art EEG technologies that can assist with minimizing TMS artifacts and noisy components. We specifically review EEG recording systems (Section 5.1) and EEG sensors (Section 5.2). To perform a TMS-EEG experiment, TMS operators ideally have real-time feedback to minimize artifacts contribution and maximize the signal-to-noise ratio of genuine brain responses. In order to achieve specific strategies to modify stimulation parameters and TMS unit settings, the gold standard would be to enable operators to properly check TMS/EEG data quality in real-time. Specifically, TMS operators should be able to obtain real-time information on the contribution of artifacts in the TMS-EEG data, and by using a tool such as the one described in [107], they would be able to perform corrective maneuvers to increase the signal-to-noise ratio of genuine TEPs. Using custom made software tools such as those described in [107] would enable: (i) the removal of TMS artifacts that mask early TEPs responses; (ii) the inspection of single-trial responses online and the detection of unwanted muscle activation or capacitive discharging artifacts; (iii) the discarding of artifact noisy channels; and (iv) a display of the time course of average TEPs according to their topographical distribution in the bi-dimensional map of the scalp.

### 5.1. EEG Recording Systems

EEG is an electrophysiological technique for measuring neuronal electrical activity. Typically, this is performed by attaching non-invasive electrodes along the scalp [108]. Brain waves recorded near the surface of the scalp tend to be weak (only 0–100 µV), which is one order smaller than the magnitude of TMS-evoked potentials and several orders smaller than TMS artifacts.

The TMS artifact waveform is strongly dependent on the recording settings of the EEG system. TMS pulses have a high rise time and contain high-frequency activity. As a result, sampling at a high rate during the digitization fully characterizes the TMS pulse and limits the stimulus artifact that is produced. Modern EEG recording systems allow for higher sampling rates by adjusting the EEG amplifier. Different studies suggest using a sampling rate of 4 kHz to ensure waveforms are clear in the EEG time series. However, recording several minutes of EEG time series at 4 kHz is resource-intensive in terms of data storage and signal processing. A number of recording circuits have been proposed to improve signal acquisition. Walker and Kimura [109] and Jakob et al. [110] used high-pass filters to reduce residual charge in an amplifier circuit. In [111,112,113,114,115], a slew-rate limiting circuit was proposed to remove voltage peaks in EEG. In [116], a sample-and-hold circuit was presented, which reduces stimulus artifact by electronically suppressing the output of the recording system. A study by Virtanen et al. [117] was the first attempt to develop a TMS-EEG compatible amplifier able to record EEGs with a voltage peak less than ±1.7 µV across each channel, and direct current (DC) potential shifts less than ±1.3 µV and ±0.5 µV at 3 ms and 10 ms, respectively. Ilmoniemi et al. [2] introduced a more optimal solution, presenting a pin-and-hold pre-amplifier block to shift the baseline from −50 µs to 2.5 ms post-TMS and prevent amplifier gain saturation throughout the EEG time series. Shortly after this, Virtanen et al. [39] proposed a solution based on an amplifier with “gain-control” and “sample-and-hold” circuits, which could ground the EEG time series for several milliseconds, immediately post stimulus. Both of these circuits were achieved by shunt circuit, protecting the amplifier from any differential input with several volts of magnitude. The sample-and-hold circuit could keep outputs locked from 50 µs at the baseline to 7 ms post-stimulus to avoid amplifier saturation [39]. This amplifier was later manufactured and commercialized by Nexstim Ltd. (Helsinki, Finland) as a 60-channel EEG system with gating periods starting at 0.05–0.1 ms (pre-pulse) and 2–20 ms (post-pulse), where the shortest gating period was 100 µs [37]. Amplifiers with adjustable sensitivity and operational range [23,60], limited slow rate [41,42], attenuator and semiconductor switches [118] were also proposed. Another proposed solution interposed a capacitor between the pre-amplifier and amplifier to eliminate electromagnetic interaction from the TMS coil [118]. The output signals of the EEG amplifier were then processed by a high-pass and anti-aliasing filter. In [41,42], an EEG amplifier with a limited low slew-rate (0.07 V/µs) was used, as well as low-gain bandwidth (200 kHz). Levkov’s solution [111] attempted to prevent amplifier blocking; however, it was only able to record frequencies up to 90Hz due to low pass filters. Bonato et al. [60] successfully used a DC amplifier to acquire EEG data during TMS experiments. Their BrainAmp MR-plus amplifier had sufficient operational range and sensitivity to preclude saturation during TMS stimulus, and they were able to conduct continuous EEG recordings without sample and hold-circuits. Levit-Binnun et al. later proposed another amplifier system, which was more dynamic and had a wider range [79].

Over the last decade, a number of amplifier circuits (see Table 2) have been made available on the market, which has facilitated improved, continuous EEG recording. Users have been able to adjust sampling rates and sensitivity in line with amplifier characteristics and study design. TMS-compatible EEG systems have been developed to record direct current (DC), also resulting in adjustable gain and dynamic ranges. Other amplifier systems have also been designed that can function within magnetic resonance (MR) scanners [36,119]. There has been growing interest in fostering knowledge on brain rhythms and rhythmic brain stimulation by combining repetitive (rhythmic) TMS with EEG recordings. Studies have shown that brain rhythms are causally indicated in cognitive functions [119].

To this end, it is also important to consider the synchronization of TMS stimulator units with TMS-EEG laboratory equipment. Two commonly used units are The Magstim Rapid and Super Rapid devices, which are suitable for TMS protocols with fixed stimulation frequencies. Another is the Magstim BiStim device, which has a modulated frequency. TMS units with fixed stimulation frequencies can be used to elicit trains of pulses, lasting several seconds; however, they are unable to implement randomized inter pulse intervals. TMS units that allow modulated frequencies deliver paired pulses with independent stimulation intensities and have an adjustable Inter-Pulse-Interval (IPI) in a fixed range. Generally, a fixed frequency is useful in therapeutic applications, whereas modulated frequency is useful for cortical investigations. In the literature, there are two free available solutions for IPI randomization [120,121]. In Figure 1, we propose a use case solution for the Magstim Rapid2 TMS Unit.

### 5.2. TMS-EEG Sensor Characteristics

EEG sensors measure the varying electrical signals created by the synchronous activity of neural cells near the surface of the brain over a period of time. Electrode impedance is a measure of the impediment to the flow of alternating current and is measured in ohms at a set frequency. The higher the impedance, the smaller the EEG signal amplitude, thereby reflecting the electrode’s ability to transfer signals at a given frequency. When the metal discs (electrodes) come into contact with the conductive gel, an ionic concentration is generated at the electrode–gel interface. This interface can be modelled to an equivalent circuit that resembles a capacitor (C), with a resistor (R) in parallel. Here, R represents the leakage resistance, whereas C mimics conductive gel behavior and reflects the characteristics of a capacitor [134]. The ionic potential gradient at the electrode–gel–skin interface introduces an electrical charge flow. The electrode–skin interface consists of a multilayer structure, where the shallow part is the epidermal layer, composed of dead cells and multiple substrates of active cells [134]. The dead cells have an insulating characteristic (much like a capacitor), which leads to ion concentration differences between concurrent potentials at the interface. In addition, other elements at the interface, including sweat glands, adipose tissue and hair follicles, are denoted by R and help facilitate current pathways. Sweat gland secretions and adipose tissue act as electrolytes between the electrode and the skin, which leads to additional capacitance [134].

Low impedance at the electrode–skin interface results in a better signal-to-noise ratio and more accurate EEG measurements. Using an electrolyte paste helps create an electrical bridge at the interface. An EEG sensor which is immersed in an electrolyte gel develops a polarized external layer when an electric current is introduced (hundreds of millivolts), and this is also influenced by the transducer metal characteristics. Impurities on EEG sensor surfaces and defective/disfigured paste can lead to large differences in electrical potentials and can introduce hundreds of microvolts of noise. A variety of transducer materials are used in EEG sensors, including gold, silver, and platinum [135]. However, it is important to note that, as reported in [54], only silver–silver chloride (Ag-AgCl) electrodes can reliably record slower EEG signals with a DC amplifier. Polarization leads to a voltage bias, and the electrode–gel interface then mimics capacitor behavior [135]. Conversely, when using non-polarizable sensors, the current would theoretically pass freely across the electrode–gel interface. The polarization of EEG sensors results in a lag (hundreds of milliseconds) in returning to the equilibrium potential point, when a TMS pulse is administered (and introduces a DC potential shift). The magnetic field generated by TMS is known to introduce electrical charges at the electrode–gel–skin interface [38,39,72]. Skin impurities, sweat glands and hair follicles at the gel–skin interface mimic the function of resistors. High capacitance ensues at the interface and decreases slowly, resulting in an initial drift in voltage and decay over time, as well as lasting offsets in the EEG signal (range from 5 milliseconds [23] to several seconds [37]). Gel–skin interface induced artifacts are due to electrical charges, which are higher in electrodes closest to the coil. When several TMS pulses are administered, EEG electrodes can heat up and pose a risk of burns to the subject’s skin. Studies have shown that each pulse can increase the temperature of Ag/AgCl electrodes by 4 °C [37]. Despite this, some studies have still reported stability in their bias potential [47,136,137]. As reported in [54], Ag/AgCl electrodes and Cl^−^ [47] electrolytic paste lead to excellent potential shifts stability, superior low frequency noise and low resistance, thus they are used for slow wave EEG activity recordings. Studies exploring commercially available Ag/AgCl electrodes reported a noise peak−to−peak variation, with silver plate electrodes, of 20 µV, in a signal window of 10 s [138]. Low impedance results in high conductivity and also prevents heating and motion. Veniero et al. [23] compared the time and amplitude of TEPs parameters between high (21–25 kΩ) and low (<3 kΩ) impedance electrodes, and found the difference to be of a factor of 2–3. Heating is directly proportional to the square of the electrode diameter [88]. Electrode rings with a cut shape, also known as C-ring or pellets, reduce heating, DC-offset and sensor motion. On the other hand, eddy-currents in the electrodes lead to electrode motion and overheating [89].

In [37], the authors successfully used plastic pellet electrodes that were coated with a thin layer of silver epoxy. Another important component in electrode polarization relates to the amount and type of electroconductive gel used, which remains an open challenge in TMS-EEG experiments. Bubbles or impurities can introduce a discontinuity factor in the conductive interface leading to drift and a drop in effect. A number of other factors should also be considered. As noted earlier, the skin at the electrode–gel interface resembles a short resistance and capacitance (RC) circuit that could otherwise disrupt EEG signals at low-frequencies [139,140,141]. Ag/AgCl electrodes transduce the ongoing voltage, as sensitive changes in the Cl^−^ concentration are affected by sweating, layers of dead cells and drying electrode gel. To minimize source artifacts, it is advised that C-ring Ag/AgCl electrodes with a large internal diameter or pin shape are used as they minimize impedance [137,140,141,142]. Electrodes with Teflon or plastic insulation and shielding wires are useful for avoiding coupled TMS artifacts. High impedance (about 20 kΩ) correlated with a slow recovery time (15–20 ms) in the EEG signal, and artifact amplitude was double that of lower impedance (0–3 kΩ) electrodes [72]. Using TMS-compatible EEG electrodes [72] and shielding wires, oriented perpendicular to the TMS coil handle, [38,72] ensures an improved TEPs signal-to-noise ratio and reduces TMS artifacts [38,72]. Using DC EEG amplifiers with high electrode impedance (>5 kΩ) has been shown to induce slow-wave DC potential shifts artifacts lasting 5–10 ms [23,143].

We propose the use of an EEG headset with no built-in electrodes and no hidden cables between the two layers of elastic fabric, in line with findings from Sekiguchi et al. [38]. They found that TMS-induced artifacts on EEGs can be reduced by re-arranging electrode wires prior to recording and ensuring the electrode wire on the EEG cap is twisted towards the input box, perpendicular to the orientation of the TMS coil handle. Studies have also shown that reference and ground electrodes near the hotspot area or above the TMS coil can induce further artifacts. Hence, we advise that the reference and ground electrodes are removed from the headset and placed on the forehead if appropriate.

## 6. A Synchronization Toolbox for TMS-EEG Laboratories: Magstim Rapid2 Use Case

A number of events, in psychological, neurophysiological and behavioral contexts, can induce changes that result in event-related potential (ERP). ERPs can be considered as a sequence of transient post-synaptic responses triggered by a specific stimulus. To detect ERPs, averaging techniques are commonly used, hence this procedure will enhance the signal-to-noise ratio [144]. A common approach in ERP studies is to simultaneously record and co-register trigger events used to synchronize EEG time series. In ERP studies, this means that each time series can be windowed around a Transistor-transistor-logic signals (TTLs) and then averaged across trials. Milliseconds of accuracy in synchronization are needed to avoid de-synchronization, with the onset of evoked responses. Many TMS paradigms require an accurate time synchronization of the TMS pulse with the EEG reading. Synchronizing TMS pulses with instantaneous brain oscillations can reduce TEPs variability across TMS pulses [145]. Generally, in the TMS-EEG context, the TMS unit yields streams of synchronization information, which is time locked with the ongoing stimulus. Using parallel or serial gates for analog or digital signals, respectively, the TMS unit can be used as a Master device to control or synchronize different Slave machines (e.g., EEG Amplifier, Navigated Brain System (NBS), electromyography (EMG) and other peripheric tools). Furthermore, the TMS unit itself can be controlled externally to achieve randomization in TMS pulse delivery (by means of a trigger box tool [120,121]). Synchronizing TMS pulses with instantaneous brain oscillations is a task that requires very high accuracy (down to milliseconds), and thus particular attention is paid to the design of the Master-Slave communication system. In Figure 1, we provide an interesting customized tool to synchronize a Magstim Rapid2 TMS unit (e.g., Master) with different TMS-EEG laboratory machines.

To easily assemble our proposed synchronization box, we note some basic required components: (i) DB26 cable with female/female connector; (ii) DB26 Slim Breakout Boards with male connector. By using a DB26 Slim Breakout and a DB26 cable, it is possible to directly connect the Magstim Rapid2. Finally, by using a BNC cable screwed with a spring clamps pin onto a DB26 Slim Breakout Boards board, it is possible to acquire the Magstim Rapid2 TTLs. Arranging these electronic components in a simple layout allows for the analogical linking to be designed. On a DB26 Slim Breakout Boards board, we use pin 19 as a ground (GND), pin 6 as Rx (receiving) and finally pin 8 as Tx (transmitting).

## 7. Maximizing Signal-To-Noise Ratio of Stereotypical TEPs Starting from Data Collection: Subject Preparation Steps

In this section, we resume important steps in subject preparation and report the state-of-the-art in EEG technologies. Skin can be scrubbed using a wooden stick topped with cotton prior to applying alcohol or a special paste [66] to ensure low electrode impedance. Alcohol or special paste soften the epithelial layers of dead skin and cotton sticks remove them. Using a minimal amount of bubble-free saline gel in the electrode reservoir avoids the “bridging” effect, overheating and an eddy current, which is also dependent on the electrode’s shape [38,54,55]. An EEG headset with free electrodes allows for reference (REF) and GND sensor repositioning [117]. EEG electrodes with a pin shape or C-ring shape with a large internal diameter, with Teflon or plastic insulation and Ag/AgCl material, ensure optimal recording performance [55]. Using electrolytic paste with Cl^−^ characteristics is important to provide an electrode–skin ionic conductor interface [47]. EEG headsets with elastic fabric and no hidden wires and no fixed sensors are also advised. Electrode wire twisting and orthogonal reorientation to the coil handle should be considered to reduce electromagnetic coupling [38]. In addition, re-orientating the coil perpendicularly in respect to the midline helps minimize peripheral muscle activation and its resulting artifacts [146]. Focal 8-shaped coils [61] and monophonic or bi-phasic units [23] can also be used.

If a bi-phasic TMS unit is used, the charging time of the capacitors should be delayed by 1000 ms to overcome recharge artifacts overlapped with TEPs within 100/200 ms. Thin foam or a customized 3D printed spacer could help bypass electrode motion or SEPs artifacts. However, it is important to consider the increased coil-to-cortex distance, which can alter electric field distributions. Customized noise masking can be administrated using ear plugs to overcome AEPs [72,77,87], while thin foam (∼0.5 mm) avoids or reduces bone conduction (SEPs) [13]. To synchronize the TMS unit and the EEG amplifier, a linking box should be used (see Figure 1). The TMS operator should try to ensure the subject is relaxed to reduce involuntary and voluntary muscle contractions. Muscle fiber activation can introduce strong noise components in the EEG time series, which can mask TEPs [147]. Finally, the navigation system should be used to ensure the repeatability of the TMS-EEG experiments and the reproducibility of TEPs. A location control tool should be a built-in solution in Neuronavigation Systems to handle TMS unit timing and help address factors including coil position, tilt and angle, and electric field orientation. All these features are implemented in the Nexstim Navigated Brain Stimulation System (NBS), which is manufactured by (Courtesy of Nexstim Ltd., Helsinki, Finland). Different TMS protocols are associated with specific stimulation parameters, including the variable/fixed inter-pulse interval. The IPI may significantly affect the direction of induced neuroplasticity and the possible interference of habituation and expectation effects. To manage pulse timing in TMS units, free solutions [120,121] may be used. For example, slow rTMS protocols (<1 Hz) often require pulses to be delivered at a stimulation rate that is randomly set around a central frequency. This jitter frequency can help prevent the occurrence of “habituation” and expectation effects. A better understanding of TMS artifacts characteristics is useful in selecting the state-of-the-art TMS-EEG devices and rapidly implementing corrective procedures, as reported in this survey. We were unable to identify any standardized protocols or frameworks; however, we propose considering the implementation of the procedures depicted to help standardize and improve TMS-EEG experiments. Customized software tools for online TEPs readout and a well-designed laboratory layout (see Figure 2) would allow for real time maneuvers to minimize artifacts contribution and concurrently maximize the signal-to-noise ratio of TEPs, starting from data recording.

## 8. Existing Challenges and Future Goals

During TMS-EEG experiments, users are often required to manually estimate optimal parameters. These include coil location, tilt and orientation, direction of induced current, intensity of stimulation, focality and spatial resolution of the magnetic field. Finding the stimulation location and setting parameters is user-dependent, varying based on the operator’s experience, which leads to the introduction of confounding factors if randomly managed with respect to the required task. The accuracy and repeatability of the current approach is therefore of concern. In addition, TMS artifacts are several orders of magnitude larger than the neural signals of interest on the EEGs readout, which at times can lead to the complete masking of early TEPs components (e.g., first 8/15 ms). Disentangling artifacts from TEPs during data analysis is a challenge and is computationally burdensome. Therefore, the ideal approach would consist of simultaneously minimizing artifacts contribution and maximizing the signal-to-noise ratio of TEPs, but doing so from the data acquisition stage. Specific strategies and training for adjusting stimulation parameters should be provided to ensure operators can assess data quality in real time (see Table 3). Software tools can also be used to help acquire high quality TEPs and provide feedback to operators in real-time. Another open challenge is related to front-end EEG technologies.

Despite technological advancements in EEG amplifiers (see Table 2) and headsets, there is still equipment in the market, which is leading to noise components and sub-optimal readings during data collection. Frameless stereotactic neuronavigation systems combined with MR images are gradually becoming the mainstay in TMS. Such systems of simulated brain navigation permit the localization of targets in a 3-D space. Localization tools allow the fine positioning of coils to within 5 mm of the MRI-defined regions of interest (ROI).

For example, the Navigated Brain Stimulation (NBS) system is a tool used for non-invasive mapping of the cerebral cortex. The stimulation location can be determined, and dimensions of the TMS-induced electrical field can be tracked in real-time, whilst accounting for factors such coil rotation, tilt and pitch. NBS provides near real-time feedback, which ensures the coil is adequately targeted and remapped during stimulus TMS [72,167], and can also obtain accurate spatial precision (sub-millimeter) during navigated TMS (nTMS) [72]. However, it is important to note that during TMS, a shift of a few millimeters or a small angular variation of the coil can significantly alter electric field distribution and penetration. This can result in an altered response, lack of a response or modifications to the plasticity effect [105,168,169]. Minimizing small shifts around the stimulation location during each pulse will enable accurate and optimal targeted stimulation. In turn, this leads to reduced variability in the amplitude of TEPs and ensures more consistent modulation of cortical excitability [72,170,171,172,173]. Hence, if the coil position deviates from the target ROI on the brain map, NBS software should temporarily block the TMS Stimulator Unit until the coil is repositioned correctly. In future, the development of multi-locus and more focal TMS coils [158], more advanced tools for online TMS artifact monitoring, [107] and real-time reduction maneuvers could increase TEPs accuracy and reproducibility and reduce operator variability.

It is important to note findings from a recent study by Conde et al. [174], which has stimulated much discussion in the literature. They concluded that TEPs can be contaminated by the effects of concurrent, non-transcranial stimulation, i.e., that studies do not currently dissociate between truly transcranial and non-transcranial components of TEPs, and they proposed the use of a peripheral multisensory control stimulation to address this. In response to this, Belardinelli et al. [175] highlighted the concern that the evoked responses obtained by the authors from real TMS and sham conditions are very different from those reported in a number of other studies, thereby limiting the generalizability of their findings. Nevertheless, such studies, including our present survey, have highlighted the need to focus on the reproducibility of TEPs across experimental setups.

Technological advancements in TMS-EEG laboratory equipment and layouts has led to the highly synchronized communication of devices and improved data collection. In this setting, complex feedback, through open or closed loops, can give further insights into the functional mechanics of cortical activity in the human brain, and has much experimental and therapeutic potential [176,177]. Capturing higher quality TMS-EEG data, with a reduction in noise, has become a slightly less arduous task, and has allowed researchers to develop novel techniques to identify and understand patterns of clinical significance [178,179,180,181,182,183,184,185]. We believe our survey highlights challenges and proposes solutions related to TMS-EEG experiments. In the current literature, most solutions are centered around the following topics: (i) standardized criteria for subject preparation (see Table 3); (ii) development of real-time artifacts minimization methods and tools for online TEPs monitoring; and finally (iii) EEG systems characteristics. The TMS-EEG community should also aim to acquire data that are comparable across laboratories, for which a standardized acquisition pipeline should be proposed. In future, it would be interesting to assess if conducting TMS-EEG experiments using an approach similar to ultrasound examinations improves data acquisition. Using an ultrasound approach, the TMS coil would be freely and continuously moved until EEG readings demonstrated adequate lateralization, latency and maximal TEPs amplitude with minimal artifact contribution.

## 9. Conclusions

In this survey, we provided a detailed review of studies in the TMS-EEG field over the last 40 years. We presented the state-of-the-art in EEG technologies, TMS-EEG artifacts, their sources and waveforms, subject preparation frameworks, and online artifacts reduction maneuvers. We discussed key examples and important features of TEPs observable in successful TMS-EEG experiments. We also noted that genuine TMS-evoked potentials are dependent on several factors that can be difficult to control, even when following best practices. To date, there is no clear position on TMS user skills in the scientific community. However, it is advisable that every operator has basic knowledge of brain physiology, the mechanisms of TMS and EEG front-end technologies, as well as TMS artifacts. In addition, training should be provided to help deal with online procedures for real time artifacts minimization and the simultaneous recognition of optimum TEPs features. In summary, this review does not provide an in-depth discussion of the training and skills required to conduct TMS studies. In contrast, we report considerations related to the lack of: (1) standardized subject preparation procedures; (2) tools for online TEPs readout and real-time artifacts correction starting from data acquisition; (3) hardware/software tools to fine target the coil in the hotspot during TMS-EEG experiments and concurrently stop the pulse delivery (TMS unit) if the coil goes out of hotspot; (4) finally, standardized EEG front-end characteristics. Despite the discussed advancements, minimizing or removing artifacts from EEG recordings remains an open challenge. The lack of standardized protocols and user’s skill is reflected in the high heterogeneity of TEPs components reported in the literature. Standardized subject preparation, tools for online TEPs readout, and standardized EEG systems are able to minimize artifacts. The TMS-EEG community should encourage greater sharing of protocols, data and tools, including TEPs readout tools for online artifacts rejection and reduction.

## Figures and Tables

**Figure 1 sensors-21-00637-f001:**
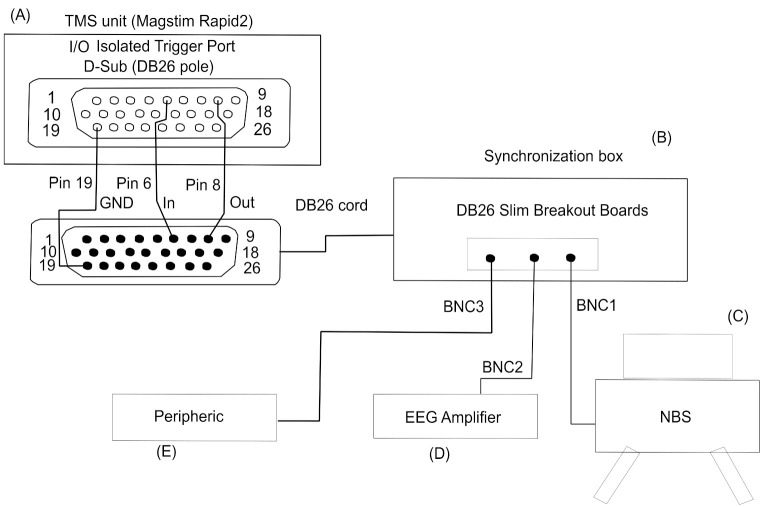
Proposed design for a customized synchronized layout for the Magstim Rapid2 TMS unit. (**A**) Magstim Rapid2 TMS unit; (**B**) our proposed Synchronization box; (**C**) the Neuro Brain Navigation System (NBS); (**D**) EEG Amplifier and (**E**) any other peripheral device (e.g., customized software suites for real-time TMS-evoked EEG potential (TEP) visualization). The Magstim Rapid2 unit streams Transistor-transistor logic data by means of a DB26 gate. Using a specific pin in the insulated I/O DB26 gate, it is possible to ensure data are synchronized with third party hardware, and concurrently the EEG recording system. Furthermore, using pin 6 in the DB26 gate, it is possible to fine tune the timing of pulse delivery with the Magstim Rapid2. For further details on the Magstim Rapid2 DB26 pinout, see the Magstim data sheet.

**Figure 2 sensors-21-00637-f002:**
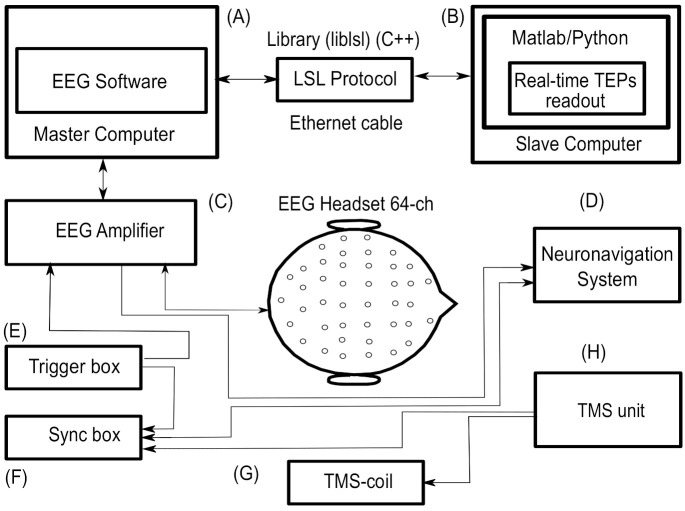
TMS-EEG laboratory layout block diagram. Our proposed TMS-EEG laboratory set-up is based on continuous “dialogue” between each device in this setting. Here, (**A**) represents the Master Laboratory PC with EEG recording software; (**B**) a Slave PC to run a customized software suite for real-time TEPs readout; (**C**) represents a TMS compatible EEG amplifier system; and (**D**) the Neuronavigation Systems. In (**E**) we illustrate a trigger device to jitter Inter-Pulse-Interval (IPI) during TMS tasks; whereas in (**F**) we illustrate a synchronization box (see Figure 1) to time neural information across the device; and (**G**) represents a TMS coil. In (**H**), the TMS unit can be used like a Master device to synchronize other tools. Generally, noise masking and trigger software are running on (**A**) or (**B**). The black arrows highlight the direction of communication, from right to left.

**Table 1 sensors-21-00637-t001:** Common artifacts in transcranial magnetic stimulation-electroencephalography (TMS-EEG) data and minimization methods.

Equipment	Artifact	Minimization Methods	References	Used in
Electrical and equipment related artifacts				
EEG sensors	Slow capacitive discharging	Impedance ≤ 5 KΩ, Cl^−^ electrolytic gel, Ag/AgCl sensors and wire orthogonal arranged to coil handle	[23,38,46,47]	[43,48,49,50]
EEG sensors	Electrode Motion	C or pin shape Ag/AgCl sensors, Cl^−^ bubbles free paste, orthogonal wire to coil handle	[23,38,51,52]	[53,54,55]
EEG sensors	Polarization	Low impedance, Cl^−^ and bubble free gel, skin scrub, shielding and twisted wires	[3,23,37,38,41,42,52,54,56,57]	[58]
EEG amplifier	Step or Pulse Artifact	Direct-coupled (DC)-Amplifier, C-shape or pellet Ag/AgCl sensor with twisted cables and orientated orthogonal to coil handle	[23,39,42,43,55,56,59,60]	[61,62,63]
TMS unit	Recharge Artifact	Recharge delay set to 1000 ms	[23,49,64]	
Muscle activation and spurious potentials evoked by the TMS coil				
EEG headset	Muscle Artifact	Thin foam, impedance ≤ 5 KΩ, %MSO adjustment and focal TMS coils	[3,13,36,38,54,65,66]	[38,60,62]
EEG headset	Blink Artifacts	Subject trained and soundproof, thin foam and online sensor Re-referencing	[13,46,63]	[49,67,68]
TMS coil	Click and Somatic Sensation	Earplugs/earmuffs, white or synthetized noises and thin foam	[3,13,69,70,71,72,73,74,75,76,77,78,79,80,81]	[82,83,84]

**Table 2 sensors-21-00637-t002:** An overview of electronic upgrades in an EEG amplifier.

Technologies	Method Proposed	References	Used in
Amplifier	High pass filter in the front-end stage	[109,110]	[39,114,122]
Amplifier	Sample-and-hold (S–H)	[2,39]	[3,52,80]
Amplifier	S–H and a grounded plane	[116]	[39,123,124]
Amplifier	Limited slow rate in preamplifier	[41,53]	[39,41,42,53,124,125,126,127,128]
Amplifier	DC-amplifiers with wide dynamic range	[23,129]	[9,37,79,83,130,131]
Amplifier	DC-amplifiers and adjustable operational range	[23,129]	[9,37,83,130]
Amplifier	EEG system magnetic resonance (MR) compatible	[74]	[37,48,50,73,119]
Amplifier	High sensitivity and operational range	[60]	[37,71,132,133]

**Table 3 sensors-21-00637-t003:** Special kinds of artifacts evoked by TMS: Strategies for online minimization.

Points to Consider	Device	Tips	References	Used in
Recording system	EEG amplifier	DC-Amplifier, adjustable dynamic range, high sensitivity and sampling rate	[60]	[37,71,132,133,148]
EEG sensors	EEG headset	C or pin shaped Ag/AgCl electrodes, Teflon insulated and shielded wires	[54,55]	[42,149]
EEG re-referencing	EEG headset	Elastic fabric, unconstrained sensors and free wires	[39]	[82,150,151]
Sensors impedance	Electrolytic gel	Bubbles free Cl^−^ gel, skin scrub, impedances (<5 kΩ)	[23,47,66,72,152,153,154,155]	
Artifact decoupling	EEG headset	Free, twisted and 90° oriented wires to coil handle	[38]	[48,49,130]
Subject comfortability	Comfortable chair	Neck resting on the back and hands on the pillow	[147]	
Electromagnetic noise	Room insulation	Equipment and room shielding, room temperature (<20 °C)	[72,81,97]	[36,72,74,77,80,97,103,104]
Coil click	Hardware/software	White or Synthetized noise, and thin layer of foam [13]		
Bone condition	Hardware	(0.5 mm) of thin foam interposed among coil and EEG sensor	[13,69,73,75]	
Equipment timing	Synchronization box	Master-Slave configuration	[41,53]	Solution Figure 1
Electric field focality	TMS coils design	8-shaped or Multi-locus coil	[61,156,157,158]	[159,160,161,162]
Confounding factor	Trigger box	Software to jitter IPI	[120,121]	[163,164]
Recharge artifact	TMS unit	Recharge delay at 1000 ms	[23]	[49,133,165]
TMS coil navigation	NBS system	Location and TMS unit control	[163,166]	

## Data Availability

The data presented in this study are available on request from the corresponding author.

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
