# Peer review of "Real-Time Artifacts Reduction during TMS-EEG Co-Registration: A Comprehensive Review on Technologies and Procedures"

_sensors, 2021, doi:10.3390/s21020637_

Round 1
Reviewer 1 Report
This is a very nice paper, that gives an extensive, up to date and accurate perspective about technique of TMS-EEG, with particular reference to the methodologies to overcome the artifacts induced by the procedures.
I've only minor concerns:
The paragraph cncerning physics of TMS is too long and in some part quite difficult to read. I suggest authors to shorthen it trying also to simplify some concepts. Particularly, I suggest to avoid mathematical formulas accross text, they could be added as a separate appendix.
More, one general picture on TEP could be added.
Author Response
Thanks for your kind review.
We tried to fully satisfy your requests. The paper has been substantially revised and reduced.
Reviewer 2 Report
The literature review is well written and informative.
Author Response
Thanks for your comments.
The paper has been sensibly revised in the present version.
Reviewer 3 Report
In the submitted manuscript, the authors provide a review of transcranial magnetic stimulation (TMS) with an emphasis of electroencephalography-based recordings of TMS-evoked responses. The authors are to be commended for reviewing a large number of studies that takes into account the majority of the field of TMS-EEG since its inception in the late 1980s. However, the manuscript is excessively long and relatively unfocused with significant organizational issues. It is also not made clear how the current work extends previous reviews focusing on dealing with TMS-EEG methodological issues including artifact identification during data acquisition and analyses (e.g., Conde et al., 2019). Some additional comments are provided meant to enhance the scientific contribution of the manuscript.
Additional Major Comments:
The literature review comes across as unfocused and relatively unsystematic both in terms of the conduction of the review and interpreting the results.
Recommend revising and refocusing the paper on methodological challenges and solutions for high quality TMS-EEG recordings rather than dedicating large sections of the paper to topics like TMS physiology that have been covered in depth in several previous reviews.
Several sections and statements that have limited information contributing to the scientific merit of the work overall.
There are many instances of inadequate language and grammar along with typos that make the manuscript difficult to follow overall.
Minor Comments:
Figure 1 lacking details and description to allow adequate interpretation of the data visualized.
Figure 2 is lacking in quality and difficulty to follow.
Several formatting issues with the included tables
Author Response
We thank the anonymous reviewer for the insightful comments.
They allowed us to modify the paper taking into account both major and minor issues raised.
The length of the paper has been very reduced.
We refocused the paper on some more relevant aspects including the artefact reduction issue.
Both figures and tables have been modified or cancelled.
A grammar revision has been carried out.
Round 2
Reviewer 3 Report
The authors have addressed several concerns from the original submission in response to the comments provided. The manuscript is reduced in length somewhat and more focused on issues associated with artifact issues associated with TMS-EEG. The added content to Section 9 adds value to the manuscript. However, several sections remain that are primarily tangents from the focus of the paper and do not contribute significantly beyond work that has been previously published (e.g. Sections 5.3 and 6). There is also limited description of the specific changes made by the authors in response to specific comments provided which makes it difficult to discern how responsive the authors were to each critique. Overall, the manuscript is improved from the final version but still requires substantial editing and revising in terms of length and focus to provide significant impact on the field.
Minor comments:
Several typos and language/grammar issues remain in the text and figures (e.g. 'Visulazation' in Fig.2).
Justified formatting in tables make them more difficult to read than necessary.
Author Response
Responses to the Editor-in-chief
The authors would like to thank and express their gratitude to the editor-in-chief, the associate editor, and the reviewers for having carefully read the paper and providing their invaluable comments, which have helped improve the quality of our manuscript.
The manuscript has now been proofread and edited by native speakers. In addition, the authors have reviewed the entire paper, taking into consideration all reviewers’ comments, and edited accordingly. The details of our changes are highlighted in the paper and point-by-point answers are provided below.
Reviewer #3
Comment 1 (Round 1)
In the submitted manuscript, the authors provide a review of transcranial magnetic stimulation (TMS) with an emphasis of electroencephalography-based recordings of TMS-evoked responses. The authors are to be commended for reviewing a large number of studies that takes into account the majority of the field of TMS-EEG since its inception in the late 1980s. However, the manuscript is excessively long and relatively unfocused with significant organizational issues. It is also not made clear how the current work extends previous reviews focusing on dealing with TMS-EEG methodological issues including artifact identification during data acquisition and analyses (e.g., Conde et al., 2019). Some additional comments are provided meant to enhance the scientific contribution of the manuscript.
Additional Major Comments:
The literature review comes across as unfocused and relatively unsystematic both in terms of the conduction of the review and interpreting the results.
Recommend revising and refocusing the paper on methodological challenges and solutions for high quality TMS-EEG recordings rather than dedicating large sections of the paper to topics like TMS physiology that have been covered in depth in several previous reviews.
Several sections and statements that have limited information contributing to the scientific merit of the work overall.
There are many instances of inadequate language and grammar along with typos that make the manuscript difficult to follow overall.
Minor Comments:
Figure 1 lacking details and description to allow adequate interpretation of the data visualized.
Figure 2 is lacking in quality and difficulty to follow.
Several formatting issues with the included tables
Answer
We appreciate the reviewer’s insightful suggestions. We would to thank and acknowledge precise and useful comments that have led us to revising and refocusing the paper on methodological challenges and solutions for high quality TMS-EEG recordings.
We have ensured that reference has been made to the study by Conde et al., along with references to studies (Belardinelli et al. 2019) which have discussed their findings further, in Section 8. We have incorporated our literature review of current surveys into our Introduction, and have presented 2-3 of the most recent and relevant surveys which we believe are of relevance to the field and our work.
We have re-edited the manuscript to review instances of inadequate language and grammar along with typos that make the manuscript difficult to follow overall. We shortened the paper by remove several sections and make our paper more focussed, as suggested by Reviewer.
Reviewer #3
Comment 1 (Round 2)
The authors have addressed several concerns from the original submission in response to the comments provided. The manuscript is reduced in length somewhat and more focused on issues associated with artifact issues associated with TMS-EEG. The added content to Section 9 adds value to the manuscript. However, several sections remain that are primarily tangents from the focus of the paper and do not contribute significantly beyond work that has been previously published (e.g. Sections 5.3 and 6). There is also limited description of the specific changes made by the authors in response to specific comments provided which makes it difficult to discern how responsive the authors were to each critique. Overall, the manuscript is improved from the final version but still requires substantial editing and revising in terms of length and focus to provide significant impact on the field.
Minor comments:
Several typos and language/grammar issues remain in the text and figures (e.g. 'Visulazation' in Fig.2).
Justified formatting in tables make them more difficult to read than necessary.
Answer
We appreciate the reviewer’s insightful suggestions and agree that these would be useful to make our paper more focussed. Moreover, we recognize that our paper at this stage was still too lengthy, and have now reduced the size from 41 pages at submission, 23 pages at present (including references). We have removed the following sections (whilst incoporating some of the information in other parts of the paper, as appropriate, for completeness and readability): i) “Literature review”, 2) “Auditory and Sensory TMS-EEG Artifacts” and 3) TMS unit and coil.
We shortened sections 3-4-5-6-7-8-9 and carefully re-edited the entire paper removing typos, extending the acronyms and condensing sentences awkward to read.
We have the previous Section 5.3 and 6.
We have modified figures and captions in an attempt to make them clearer.
We have recast tables 1 and 2.
We have highlighted some of the relevant changes by changing text colour to blue.
We would like to conclude by highlighting some of our key contributions, including proposed customizable tools and laboratory layout, for device synchronization. We are also one of the few studies to discuss approaching TMS as we do ultrasounds, and associated future work relating to this. We also re-emphasize the importance of the operator’s knowledge and manual skills and how this can affect the quality of the TEPs.
